psychology/computer graphics

virtual reality, reinforcement learning, factorial design, participant preference, walking-in-place, rendering quality

**Author for correspondence:**
Mel Slater
e-mail: melslater@ub.edu

†Present address: Artanim Foundation, 40, Chemin du Grand-Puits, 1217 – Meyrin (Geneva), Switzerland.

# Evaluating participant responses to a virtual reality experience using reinforcement learning

Joan Llobera[1,†], Alejandro Beacco[1], Ramon Oliva[1], Gizem Şenel[1,2], Domna Banakou[1,2] and Mel Slater[1,2]

[1]Event Laboratory, Department of Clinical Psychology and Psychobiology, University of Barcelona, Barcelona, Spain
[2]Institute of Neurosciences of the University of Barcelona, Barcelona, Spain

JL, 0000-0002-9471-1334; AB, 0000-0001-8192-1431;
RO, 0000-0002-6472-8573; GŞ, 0000-0003-3750-6964;
DB, 0000-0002-0974-6971; MS, 0000-0002-6223-0050

Virtual reality applications depend on multiple factors, for example, quality of rendering, responsiveness, and interfaces. In order to evaluate the relative contributions of different factors to quality of experience, post-exposure questionnaires are typically used. Questionnaires are problematic as the questions can frame how participants think about their experience and cannot easily take account of non-additivity among the various factors. Traditional experimental design can incorporate non-additivity but with a large factorial design table beyond two factors. Here, we extend a previous method by introducing a reinforcement learning (RL) agent that proposes possible changes to factor levels during the exposure and requires the participant to either accept these or not. Eventually, the RL converges on a policy where no further proposed changes are accepted. An experiment was carried out with 20 participants where four binary factors were considered. A consistent configuration of factors emerged where participants preferred to use a teleportation technique for navigation (compared to walking-in-place), a full-body representation (rather than hands only), the responsiveness of virtual human characters (compared to being ignored) and realistic compared to cartoon rendering. We propose this new method to evaluate participant choices and discuss various extensions.

## 1. Introduction

The construction of virtual reality (VR) applications is a complex task involving the choice of appropriate hardware and

affordances provided by the software and remains so after three decades of research and deployment of applications in this field. Typically, a VR application requires motion capture of actors, scanning of environments, a team of designers, artists and so on. Moreover, VR is rapidly moving towards becoming a major consumer product, with high-quality head-mounted display (HMD) systems available at costs cheaper than many smartphones and with millions of sales throughout the world. Application designers face a number of trade-offs—for example, should the environment look as photorealistic as possible, or be more cartoon-like? The virtual space in which people can move will typically be much larger than the physical space in which they operate, so an interface method is required to enable movement around the virtual space. Should this interface method stress realism or efficiency? Moreover, choices among these and many other aspects of the final application will have implications for the quality of experience of participants. Although one choice may be, from an engineering or monetary point of view, more expensive than another, it may considerably enhance the quality of experience. Moreover, these individual factors are unlikely to operate in isolation from one another but taken together will contribute to an overall quality of experience that is more than just the sum of the individual components. However, how should the quality of experience of participants in a VR be specified and measured? This is normally accomplished through questionnaires, especially around the concept of 'presence', that is, the sense of 'being there' in the place depicted by the computer displays.

The concept of presence has a long history going back to the notion of telepresence introduced by Minsky [1]. When a human operator perceives and acts through a remote teleoperator system, they typically have a sensation of being at the remote location. Following the first burst of interest in VR that started in the late 1980s, this same concept was used to portray the sense of being in the place depicted by the virtual environment [2,3]. Interest focussed on those factors that might contribute to that feeling [4–6], and a number of specific experimental studies to try to elicit these were carried out—e.g. display update rate [7], type of interaction and display parameters [8,9], pictorial realism [10], the influence of psychological factors such as field dominance [11], the role of self-representation with a virtual body [12,13], the influence of body movement [14] or the influence of illumination realism such as dynamic shadows [15]. This type of research continued through the 1990s, in spite of the problem that presence was ill-defined and measured through questionnaires.

Meehan *et al.* [16] operationalized the idea that if presence occurred then people should respond physiologically to the threat. For example, when participants stand close to a precipice in VR their heart rates typically increase. This method was later used to study how variations in end-to-end system latency influence presence (as operationalized through heart rate changes), showing that lower latency systems produce a greater average increase in heart rate in response to the precipice than higher latency systems [17]. This work on presence is substantially reviewed in [18] where it was argued that operationally presence is indicated when participants can be observed to respond realistically to events and situations in the virtual environment. An illustration would be, for example, that when talking to a virtual audience, people with a fear of public speaking should display anxiety, and this does happen—e.g. [19,20]—even though all participants know that there is no actual audience there. In fact one of the greatest uses of VR, exploiting this presence-inducing property, has been in the field of clinical psychology—a huge amount of work on anxiety disorders [21], post-traumatic stress disorders [22] and even on psychosis, including paranoia—e.g. [23–25]. However, the problem of measurement of presence has remained outstanding. Reliance solely on questionnaires is unconvincing—see [26].

To some extent, the measurement of presence is analogous to measurement in colour perception. The VR system provides technical capabilities (essentially the physics, that which is actually supported using the system) and presence is a subjective correlate. Similarly, an object reflects or emits light at different wavelengths (the physics) and the human perceptual system interprets the result as colour (the subjective response). In [27], we exploited this analogy by introducing a methodology for the measurement of presence based on colour matching theory, which did not require questionnaires at all. In this method, participants were introduced to a number of factors in the VR that they could modify during the course of their exposure—field of view, whether or not they had a virtual body that moved synchronously with their own movements, whether they saw the virtual environment from a first- or third-person perspective, and quality of visual rendering. The factor levels were ordered from 'low' to 'high'—for example, the field-of-view could be the actual field-of-view of the HMD used (high), or 60% of that (low); there might be no virtual body representing the participant (low), a static one (medium) or one that moved in synchrony with the participant movements (high). They were first exposed to a virtual environment with all factors at the highest level possible. Half of the participants were asked to pay attention to place illusion (PI; the illusion of being in the space)

and the other half to plausibility (Psi; the illusion that the virtual events were really occurring) [28]. Thus, participants were asked to concentrate on one of these two sensations, which we refer to as the 'target'. They were then trained on each of the factors that they could modify, and after this training, they were exposed to a virtual environment with all factors at a low level. At various moments during the exposure, they were asked if they wanted to enhance any one of the factors by one level. They were told that they should stop changing the factor levels whenever they felt that they had matched their earlier target sensation of presence (either PI or Psi). The number of changes they made was subject to an economic constraint.

This method does not require the use of questionnaires, it only relies on a 'match' between sensations, much like in colorimetry, where participants are never asked, for example, 'How much do you experience this colour as red?' but are only asked about their sensation of a match between a target patch of colour and a projected patch of colour that they manipulate themselves through red, green and blue projectors. Of course, the two major differences between the colorimetry method and the presence method are that the matches are between sensations at two different times (rather than comparing two colours simultaneously) and the comparison of subjective presence is more complex than comparison of colours. However, the method did produce different results depending on whether participants had been asked to pay attention to their sensation of PI or their sensation of Psi, and the results were theoretically sensible.

This method has been used now several times in other research. In [29], the method was closely followed and augmented with EEG measures of engagement, and in [30], it was applied to the evaluation of auditory environments. In [31], it was used to evaluate participant Psi responses to four different factors influencing the portrayal of a string quartet performance in VR, and in [32] to evaluate the influence of virtual human character behaviour on Psi. In [33], the method was used to assess the believability of a virtual environment that simulated rock climbing. In [34], the method was used to assess factors that contributed towards the sense of embodiment of humans in avatars. In [35], the method was used to evaluate an avatar giving a TED-style talk to a participant. In this case, the stopping rule was not presence, believability or the sense of embodiment, but rather just participant preference. The matching criterion was that the participant had reached a configuration of factors where they did not require more changes in order to be satisfied, irrespective of their reason for this. In this case, they did not have to experience a first exposure where all factors were set at a maximum level, and then later compare their current sensation to that earlier exposure.

A common feature of all previous uses of this method is that participants were presented with moments when they could choose to make a change to their current configuration and thereby make a transition. The transitions chosen were up to them, under various constraints depending on the particular application.

In this paper, we present a new approach whereby there is an underlying reinforcement learning (RL) agent that presents binary options for a change on one particular factor at a time to participants at regular time intervals. Instead of focussing on presence, following [35], the participant was asked to select changes to factor levels based solely on their preference, although, here they were only required to accept or reject a change proposed by the RL agent. The sequence of possible changes was initially chosen by the RL at random. Based on the acceptances and rejections of the changes offered, the RL agent learned which of the options that it could offer would be most likely to be accepted. In other words, the RL agent formed a 'policy': given the state of its environment (the current configuration of factors), it estimated the probability of a certain outcome associated with each of the possible options it could offer. Its goal was to maximize its 'reward', which in this case was based on whether the participant accepted the change it proposed. Convergence of the RL was indicated when the participant continued to reject all changes.

Hence there are, therefore, two important differences with previous work. The first is that we do not impose the criteria of presence on participants, they are free to select changes to the factors in accordance with their preferences, for whatever reason. The second, and most important difference, is that proposals for change of factor levels are chosen by the RL, which therefore introduces a clear stopping rule—when no more changes are accepted by the participant—indicating convergence of the RL, in the sense that additional changes would not improve participant preference.

We demonstrate this method in an experiment involving four binary factors. Although the choice of factors was motivated by previous results in the presence literature, it is important to note that presence (whether PI or Psi) was never mentioned to participants as a criterion for selection between alternate choices offered by the RL—this was strictly based on participant preference. In a traditional factorial experiment, four binary factors would require a $2^4$ factorial design in order to fully test all

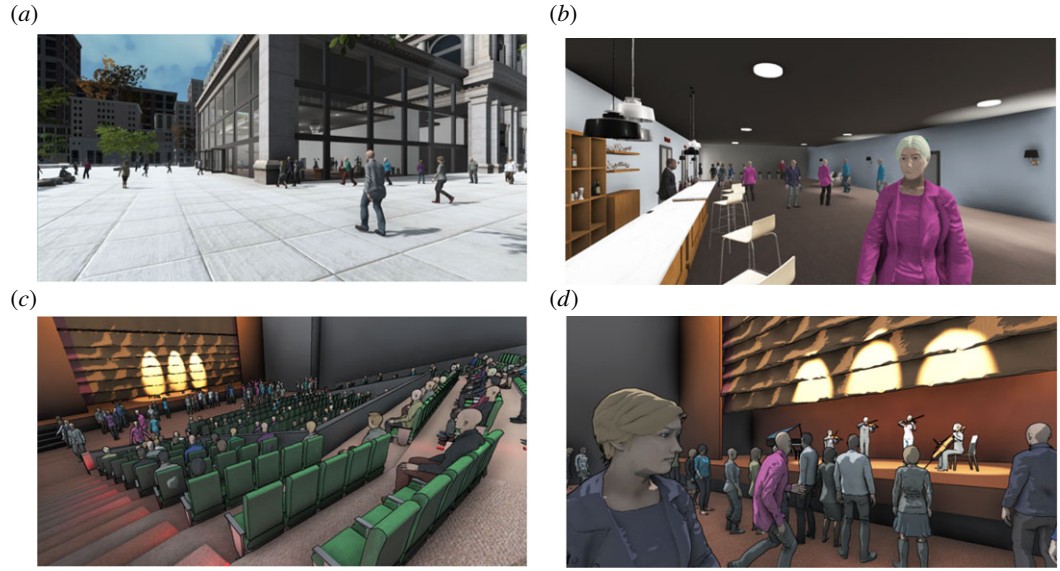

**Figure 1.** The scenario (*a*) the scene outside of the building showing the crowd with realistic rendering. (*b*) Inside the theatre building, notice one of the characters is looking towards the participant, and with realistic rendering. (*c*) In the theatre going down the steps, with cartoon rendering. (*d*) The concert itself but seeing the stage from the viewpoint of the participant with the audience around and cartoon rendering and the nearby character is not looking towards the participant.

combinations of the influence of these factors. For a within-group study, this implies requiring each participant to do a task in VR and then answer a post-experimental questionnaire for each combination of experimental factors, i.e. 16 times. In a between-group study, each participant would experience one combination of factor levels, but a much larger sample size would be needed in comparison to within-groups. Here, we show the RL-based method can work with only 20 participants, in this particular study. The RL-based method uses a straightforward technique (Q-learning) because the experimental design involved a set of factors that does not require masses of data. Contemporary deep RL algorithms such as proximal optimal policy algorithms [36] or the soft-actor-critic [37] can converge for more complex experimental designs, at the price of needing significantly larger samples.

The task of participants was to make their way in VR from a street scene to inside a concert hall and then find their way into the theatre to observe part of a concert. This was motived by the overall project of which this research is a part, which is concerned with attendance at concerts in VR, for example [38].

Participants on entering the scene were in a street by a concert hall building, among a crowd of virtual people. The rendering and some interactive aspects of the scene were determined by four factors, and for each we cite previous work where this factor or similar has been studied with respect to presence: (i) the navigation method (walking-in-place (WiP)) where participants mimicked real walking but stayed physically in the same spot, or a point-and-click method [39]; (ii) body representation—whether participants had a full first-person perspective virtual body or just saw virtual hands [27]; (iii) social feedback—whether other crowd members would acknowledge them or not [32]; and (iv) rendering type—whether the environment was rendered more realistically or more cartoon-like [40]. Figure 1 shows the scenario, and it is also shown in the electronic supplementary material, video S1.

After a period of training where participants learned about these four factors, they made their way to the concert, and every so often the RL proposed a binary choice to either change the level of one of the four factors or leave the current configuration unchanged.

# 2. Material and methods

## 2.1. Materials

The main equipment used was an HTC VIVE PRO wide field-of-view head-tracked stereo HMD. This displays a three-dimensional scene in stereo with an approximately 110° field of view with a

resolution of $2880 \times 1600$ pixels per eye displayed at 90 Hz, together with the VIVE wireless adapter so that the HMD was untethered. The tracking was done with the lighthouse trackers provided by the HTC VIVE Pro,[1] with a resolution of a millimetre and a refresh rate of at least 120 Hz.[2] In addition to the two hand wands provided with the HTC VIVE PRO, three trackers from VIVE were attached to the participants' waist and feet to support real-time whole-body tracking. The computer used for the experiment was an Intel Core i7-8700 K@3.7 GHz, GeForce RTX 2070 8GB GDDR6 and 16GB of RAM DDR4.

The virtual environment was implemented using UNITY3D, v. 2018.2.20 and Unity's OPENVR (Desktop) package v. 1.0.5. The three-dimensional assets of the street scene were from POLYPIXEL, the concert hall and the trees in the tutorial phase were a combination of assets from the Unity Asset Store[3] and custom-built. The crowd was generated using three-dimensional characters made with Adobe's FUSE. The concert itself was incorporated from a previous study [31].

To simulate the movement of the crowd through the environment, the Unity navigation mesh was used for high-level planning and the RVO2 library [41] (http://gamma.cs.unc.edu/RVO2/) for local steering and collision avoidance.

To model the behaviour of virtual human characters, we customized scripts implementing specific rules for the requirements. For example, in the case of the concert, characters without a ticket needed to buy one, so they would go to the ticket queue, wait for their turn and buy one. Once they had a ticket, or if they already had one, they entered the theatre. To define these tasks, we used goal points, which are reached by navigating through the environment.

Animation of the characters in the crowd was carried out with Unity's animation system MECANIM. For specific and eventual tasks, attached to a behavioural decision (for example, reacting to the beginning of a concert), a corresponding animation was simply triggered, with some randomness added to provoke a variety of behaviour. For the navigation, and particularly the steering, we used blend trees to blend between several existing animations.

## 2.2. The factors and configurations

There were four binary experimental factors, the levels of which are coded as 0 or 1.

### 2.2.1. Navigation

In order to move through the virtual world, the participant could navigate using a *WiP* (1) locomotion technique [39,42–44] or a *teleportation* (0) technique using a hand-held controller. WiP, originally described in [44], requires the participant to make walking movements (lifting each leg in turn) while nevertheless staying in the same physical place. The computer program moves participants forward in the VR in the direction normal to the plane of their hips while these movements are detected. WiP uses information from three sensors (hips, left foot and right foot), averaging over the motion of the last five frames.

The alternative was teleportation, where participants used their hand-held controller to point towards a spot on the virtual ground while holding a button on the controller that then displayed a ray from their pointing hand towards the spot. When the button was released they would instantaneously teleport to that position. WiP has been associated with a greater illusion of walking and higher presence, whereas teleportation is more efficient and takes less effort but can be disorienting [39].

### 2.2.2. Body representation

When participants wearing the HMD looked towards their own body, they would see a virtual body coincident with the real body that moved in synchrony and correspondence with their real movements. In this case, participants had a choice between seeing a *full-body* representation (1) or only seeing virtual *hands* (0). In both cases, the movement always matched the participant's movements, in real-time.

### 2.2.3. Social feedback

Participants were surrounded by a crowd of virtual people who were also making their way to the concert. Social feedback refers to how the virtual crowd members responded to the presence of the

---

[1]https://www.vive.com/mx/product/vive-pro/.

[2]http://doc-ok.org/?p=1478.

[3]Particularly, https://assetstore.unity.com/packages/3d/environments/urban/urban-city-pack-34832.

participant. Responsiveness of the virtual environment to the presence of the participant has been argued to be an important contributor to Psi [28]. In this case, virtual characters responded or not to the participant. When participants looked towards a virtual human character, the character looked back at them and provided subtle social *feedback* (1) including a head nod or hand waving. Alternatively, virtual characters *ignored* (0) participants and did not respond when the participant gazed at them.

### 2.2.4. Rendering

The rendering type was either a *cartoon*-style rendering (1) or a *realistic* rendering (0) method. In the cartoon method, we used a *cel shading* effect on the fragment shader, based on an online tutorial.[4] Outlines were added on postprocessing, after rendering the scene. This gives the appearance that the scene has been drawn by hand, as if it were a cartoon or a comic book. In the realistic method, we use Unity's standard render pipeline and physically based shading for the lighting computation. In both cases, light baking and ambient occlusion were used in order to improve the visual quality of the scene (figure 1*a*,*b*).

### 2.2.5. Configurations

In total, there were 16 possible *configurations* of factors as shown in table 1.

## 2.3. Participants

The experiment was advertised to members of a database who had requested to be informed of new studies, and it was also advertised through the campus. A total of 20 people, with mean ± s.d. age 34 ± 8.7, including 13 women and seven men, participated in the study. The electronic supplementary material, table S1 gives the demographic characteristics of the sample.

Upon arrival to the laboratory, participants were assigned an identifier (ID) number. All subsequent data gathered was stored with respect to this ID, and no document except the informed consent contained both the participant's ID number and their name. Then, after signing the consent form, they completed the three main phases of the experiment. Each participant received 10 euros in the form of a gift voucher as compensation after completion of the third stage.

## 2.4. Procedures

There were three phases of the experiment: tutorial, main and confirmation. We describe each in turn.

### 2.4.1. Tutorial phase

In the tutorial phase, participants were equipped with the trackers and the HMD, given instructions, and then the VR experience began. Participants were located in a simple scene which contained only some trees. They did not have any particular goal, except free exploration.

In this phase, each experimental factor was introduced successively, illustrating its two options. Participants had to explore both options and press a confirmatory button when they understood both options. For this, they had a virtual panel, somewhat like a large digital watch, attached to the location of their left hand where they were able to choose to switch between the two levels of a factor. For example, with respect to body representation, they could switch between two buttons to portray the hands only representation or the full-body representation. On selection of one of the two buttons, the display would change accordingly so that participants could switch between the two alternatives and explore their meanings. At the end of the trial period for each factor, they could select the button 'I understand both options' (figure 2*a*).[5] They would then move to the next factor. The order of factors presented was body representation, rendering, social feedback and navigation. Participants could obtain clarification from the experimenter at any time.

[4]https://www.gamasutra.com/blogs/DavidLeon/20170519/298374/NextGen_Cel_Shading_in_Unity_56.php.

[5]All the information was presented in Spanish, we show here the translated version to facilitate understanding.

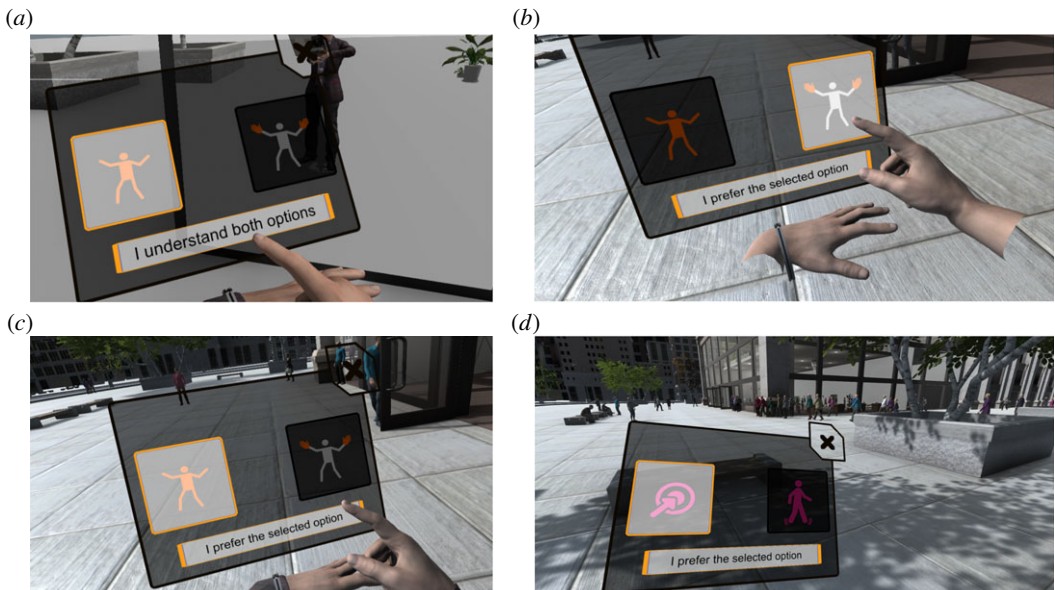

**Figure 2.** The panels were used to select the participant preferences. (*a*) During the training phase, the participant confirms understanding of both having a full body or having only hands. (*b*) During the experiment, the participant confirms selection of the hands only option. (*c*) During the experiment, the participant confirms the full body option. (*d*) During the experiment the participant confirms the teleportation option.

**Table 1.** The 16 configurations and their codes.

| no. | configuration | navigation | body rep. | social | rendering |
|---|---|---|---|---|---|
| 0 | teleportation - hands - ignored - realistic | 0 | 0 | 0 | 0 |
| 1 | teleportation - hands - ignored - cartoon | 0 | 0 | 0 | 1 |
| 2 | teleportation - hands - feedback - realistic | 0 | 0 | 1 | 0 |
| 3 | teleportation - hands - feedback - cartoon | 0 | 0 | 1 | 1 |
| 4 | teleportation - body - ignored - realistic | 0 | 1 | 0 | 0 |
| 5 | teleportation - body - ignored - cartoon | 0 | 1 | 0 | 1 |
| 6 | teleportation - body - feedback - realistic | 0 | 1 | 1 | 0 |
| 7 | teleportation - body - feedback - cartoon | 0 | 1 | 1 | 1 |
| 8 | WiP - hands - ignored - realistic | 1 | 0 | 0 | 0 |
| 9 | WiP - hands - ignored - cartoon | 1 | 0 | 0 | 1 |
| 10 | WiP - hands - feedback - realistic | 1 | 0 | 1 | 0 |
| 11 | WiP - hands - feedback - cartoon | 1 | 0 | 1 | 1 |
| 12 | WiP - body - ignored - realistic | 1 | 1 | 0 | 0 |
| 13 | WiP - body - ignored - cartoon | 1 | 1 | 0 | 1 |
| 14 | WiP - body - feedback - realistic | 1 | 1 | 1 | 0 |
| 15 | WiP - body - feedback - cartoon | 1 | 1 | 1 | 1 |

### 2.4.2. The main phase

Once the tutorial was completed, participants were transferred to a virtual street, where over 80 animated virtual human characters were walking towards a concert hall. The participant's task was to go into the concert building, and then find their way into the theatre. Once in the theatre, they had to go down some stairs and stand in front of the stage with over 120 virtual human characters. Then, the concert started which consisted of a quartet playing. After 20 s of the concert, the environment faded to black, leaving the participant back in the street again, and the same task had to be repeated: go and attend the concert.

| question: which you prefer ... | score 0 | score 10 |
|---|---|---|
| ... the world painted in a realistic way or cartoon-like? | realistic | cartoon |
| ... the configuration with more passive characters or more active? | passive | active |
| ... the configuration without a body or with a body? | no body | with body |
| ... navigation with tele-transport or imitating real walking? | teletransport | walking |

**Figure 3.** The post-experience questionnaire used the same icons to illustrate the choices as during the VR exposure. Participants were required to score their response to each question on a scale of 0 to 10.

During this task, the interface presented choices to the participants using the method presented during the tutorial, except that the button 'I understand both options' (figure 2a) was replaced by 'I prefer the selected option' (figure 2b). The factor to display was selected by the RL agent (see additional details below). Participants could choose to leave the configuration unchanged or accept the change proposed by the RL agent. Before making their choice participants could freely explore the two options presented by switching between the configuration buttons and finally had to confirm that the option currently selected was their preferred one (figure 2c,d). Once the preferred option was confirmed the panel disappeared. Eight seconds after closing, the panel would open again with a new change proposed. The whole process was repeated until participants had been offered a total of 24 opportunities to make a choice.

The only criterion given to the participants about whether they would accept a proposed change or not was their preference. This was emphasized by the wording in the interface, as mentioned, 'I prefer the selected option'.

### 2.4.3. The confirmation phase

After the VR exposure ended participants were asked to remove the HMD, and the *confirmation* phase started. First, they were asked to complete a demographics questionnaire. This obtained information on age, sex, knowledge of computing, experience with VR and computer games.

After completing this, they were requested to put on the HMD again to re-enter the environment, but for free exploration of the same scenario rather than for any specific task. They were verbally asked which configuration they wanted to have. This was asked one factor at a time: do you prefer to have a body or only hands? Do you prefer realistic or cartoon rendering? Do you prefer more interactive or more passive characters? Do you prefer to teleport or to walk in place? This was carried out as a check for consistency with the configuration they had chosen during the main exposure. After a period of free exploration of approximately 5 min, they removed the HMD.

Then, they completed a post-experimental questionnaire as a further consistency check. It was presented as an online form to check for each experimental factor, which option they preferred, on a scale between 0 and 10. For example, it asked *What do you prefer, the world painted in a realistic way, or cartoon-like?* Near to the 0 grade the icon for realistic rendering would appear, and near the 10 grade the icon associated with cartoon rendering. After each question, the online form would ask *Why?*, and the participant could provide a free written response. Figure 3 shows the icons used for each question. Notice again that the only criterion offered to the participants was their preference.

Overall, the entire procedure lasted between 40 and 50 min, with the VR exposure lasting approximately 15 min. The exact total duration changed among participants because they spent most of the time exploring the VR environment and the options among which to choose, and therefore, there was a variable delay between when a new choice was presented to the participants, and when they completed the choice.

## 2.5. Design

One aspect that was crucial for a successful assessment of participant preferences within a large variety of configurations was to design a task that appeared simple enough to the participants. In this case, there were three decisions that were adopted to facilitate the task from the participant's perspective.

First, the proposal of a configuration change always showed the options available in the same format and location. In this experiment, the current configuration was always on the left, and the change proposed was always on the right side of the panel. To actually explore the configuration change proposed, the participants had to explicitly select the appropriate icon. The configuration only changed when the participant actively selected the configuration change proposed.

Second, before giving a response, the participants were free to explore the current configuration at their own pace, as well as the new configuration, to go back and forth as many times as wanted, and explore the environment with either option activated. Pilot studies showed that this was crucial for the participants to feel comfortable with the task and to carefully explore the options when their preference was not clear to themselves. To facilitate the association of a button with an experimental factor, we designed a small set of icons, which were used throughout, including for the questionnaire, as shown in figure 3.

Finally, to avoid confusion, the configuration in which the participant started the main experiment followed on from the configuration resulting from the trials performed during the tutorial phase. Therefore, when starting the experiment, they were already aware of the current configuration—including the active navigation technique, so that they were immediately able to do the task. This also meant that each participant started with a different configuration, and this configuration was not chosen randomly but was the result of their actions during the tutorial phase. This method was adopted as a result of extensive piloting, where we learned that there was disorientation if, after the training period, the configuration were reset to another one than the one that they happened to be experiencing at that time. Nevertheless, participants then went on to make multiple configuration changes.

### 2.5.1. Reinforcement learning

Each experimental configuration (table 1) corresponds to a *state* of the RL algorithm. In particular, we used Q-learning (introduced in [45] summarized in [46] and in [47], chapter 6), adapting an example available online (https://github.com/Unity-Technologies/Q-GridWorld). In RL, the algorithm attempts to maximize the long-term reward to the agent through appropriate choices of actions $a$, in the context of a state $s$. The specificity of Q-learning is that it uses a table called Q-table, which contains an estimated value of each action $a$, for a given state $s$. To update the values of the Q-table, for $s$ as the current state and $a$ the action of the agent, the formula used is

$$Q[s,a] = l(r + \gamma \times \max(Q(s_p))) + (1 - l)Q[s,a]. \tag{2.1}$$

In equation (2.1), in addition to an action $a$ and a state $s$, we normally consider:

— a reward value $r$, which is a value, generally bounded between −1 and +1, that the agent receives in response to the action taken;
— $\max(Q(s_p))$ is the maximum value of the row in the Q-table corresponding to the proposed state. It is here used as an estimate of future long-term rewards;
— a discount factor $\gamma$ which represents the weight given to (estimated) possible future rewards. For stability reasons, it must always be smaller than 1; and
— a learning rate $l$ which represents the speed at which the estimated values update. It is a number smaller than 1 and usually close to 0.

In this particular case, the goal of the algorithm was to find the changes in the configuration of the VR that a participant *was most likely to accept*, among a set of possible options. Therefore, the state $s$ corresponded to the current experimental configuration of the VR, the action $a$ corresponded to a change in experimental configuration that the RL algorithm proposed, and the reward $r$ was determined by whether the participant accepted the change proposed by the RL algorithm or not (+1 or −1, respectively). When the reward was received, the position of the Q-table corresponding to state $s$ and action $a$ was updated according to equation (2.1).

In addition, because it was likely that different participants might have different preferences, we chose to run the RL algorithm separately for each participant.

Overall, since the experiment was quite short for RL standards, and therefore the system had little time to estimate long-term rewards, we chose a rather small discount factor and a moderate learning rate. These parameters took the following values: $l = 0.2$, for the learning rate; $\gamma = 0.15$, as the discount factor. All values in the Q-table were initialized to 0.

In order to summarize the preference estimation performed by the RL algorithm, we used the values in the Q-tables, averaging across participants. We were interested in determining the configurations in which participants most liked to stay, i.e. the configurations in which they were most likely to reject transitions to another configuration. Because we had a separate Q-table for each participant, we required a measure to combine these results. We therefore derived a score to summarize these data. Let $Q_p[s, a]$ be the Q-table for the $p$th participant, where $s$ is the configuration and $a$ is a proposed change in configuration. Then, we define

$$\text{RLscore}(s) = \sum_p \sum_a Q_p[s, a]. \tag{2.2}$$

This is the sum of all values over all actions and participants and gives us a value associated with each RL state ($s$). The lower this value, the higher the tendency for participants to reject changes when in that configuration. Therefore, the more participants are inclined to stay in $s$ the lower the value of RLscore($s$).

As complementary measures, to better understand the RLscore, we also use the number of visits, i.e. the number of times participants landed in one configuration (over a total of 24 choices × 20 participants = 480 visits), and the stay ratio, i.e. the frequency at which participants decided to stay in a given configuration, given the chance to change.

# 3. Results

## 3.1. Comparison of configurations

Table 2 shows the experimental configurations sorted by RLscore showing that by far the preferred configuration was 0110 (navigation by teleportation, body representation with the full body, social with feedback and rendering as realistic). The number of visits is also a clear indicator of the options explicitly preferred by the participants: the most visited configuration was the one participants most often chose as their preferred option when offered to return to the environment for free exploration after the main part of the experiment (13 out of 20). The five most preferred configurations were also the ones with most visits, and the ranking configurations according to reported preferences fit with the ranking of configurations according to the number of visits. The configuration 0110 was also the one with the highest staying ratio (86%). The staying ratio was also higher for the five configurations preferred by participants, although the order is not preserved. A graphical representation of these results is available as the electronic supplementary material, figure S1.

## 3.2. Statistical analysis

It is possible, though unlikely given the results of table 2, that participants accepted or rejected proposals for changes randomly rather than actually according to their preferences. Each participant made 24 choices distributed into the 16 configurations shown in table 1. Let $p_j \geq 0$, $j = 0, 1, \ldots, 15$ be the probability of being in configuration $j$ after a proposal by the RL. Then, $\sum_{j=0}^{15} p_j = 1$ and the situation corresponds to a multinomial distribution with $N = 24$. We have $n = 20$ observations on the random vector $r_i = (r_{i0}, r_{i1}, \ldots, r_{i,15})$, $i = 1, 2, \ldots, n$ corresponding to the number of times the $i$th participant was in each of the configurations, i.e. $r_{ij}$ is the number of times that participant $i$ was in configuration $j$ after a proposal. These correspond to 'no. of visits' in table 2.

We let the prior probability distribution of the $p_j$ be jointly uniform, which corresponds to a Dirichlet distribution with all parameters 1. The likelihood is the multinomial distribution as described above. We used the Stan system (https://mc-stan.org) [48,49] to derive the posterior distributions of the $p_j$. This was carried out with 2000 iterations and four chains and achieved convergence without problem. All Rhat = 1 indicating the results of the four independent simulation chains converged and mixed.

The posterior distributions, created using the Bayesplot [50] library, are shown in figure 4. The posterior distribution of configuration 6, 0110 (teleportation, body, feedback and realistic) stands out as having the

**Table 2.** The different configurations sorted by RLscore. (*visits* is the number of visits across subjects (from a total of $24 \times 20$ = 480). *stay ratio* is the proportion of configuration changes proposed and rejected when in that configuration. *RLscore* estimates the value that the RL algorithm associates with leaving that configuration (the lower the more likely it was that people stayed in that configuration). *post. pref.* indicates the post-experimental preference, i.e. the number of participants out of 20 that selected that configuration as their preferred configuration during the confirmation phase. *last visit* shows the number of participants whose last visit corresponded to the given configuration. The configurations are shown in table 1.)

| no. | configuration | no of visits | stay ratio | RLscore | post. pref. | last visit |
|-----|---------------|--------------|------------|---------|-------------|------------|
| 6 | 0110 | 168 | 0.82 | −9.54 | 13 | 9 |
| 14 | 1110 | 72 | 0.75 | −3.00 | 2 | 2 |
| 2 | 0010 | 53 | 0.72 | −2.37 | 2 | 2 |
| 4 | 0100 | 49 | 0.67 | −2.12 | 2 | 5 |
| 0 | 0000 | 42 | 0.81 | −1.85 | 1 | 0 |
| 5 | 0101 | 14 | 0.71 | −0.55 | 0 | 0 |
| 10 | 1010 | 14 | 0.64 | −0.33 | 0 | 1 |
| 13 | 1101 | 7 | 0.57 | −0.11 | 0 | 0 |
| 3 | 0011 | 9 | 0.56 | −0.09 | 0 | 0 |
| 8 | 1000 | 14 | 0.5 | −0.02 | 0 | 0 |
| 9 | 1001 | 4 | 0.25 | 0.00 | 0 | 1 |
| 1 | 0001 | 2 | 0.5 | 0.00 | 0 | 0 |
| 11 | 1011 | 3 | 0.33 | 0.10 | 0 | 0 |
| 15 | 1111 | 10 | 0.4 | 0.19 | 0 | 0 |
| 7 | 0111 | 6 | 0.33 | 0.20 | 0 | 0 |
| 12 | 1100 | 13 | 0.38 | 0.25 | 0 | 0 |

**Table 3.** Summaries of the posterior distributions of the factor level probabilities showing the mean, standard deviation and 95% credible interval for each.

| factor level (1) | mean | s.d. | lower quartile | upper quartile | 1 - mean | factor level (0) |
|------------------|------|------|----------------|----------------|----------|------------------|
| WiP | 0.29 | 0.02 | 0.25 | 0.33 | 0.71 | teleportation |
| body | 0.70 | 0.02 | 0.66 | 0.74 | 0.30 | hands |
| feedback | 0.69 | 0.02 | 0.65 | 0.73 | 0.31 | ignore |
| cartoon | 0.13 | 0.02 | 0.10 | 0.16 | 0.87 | realistic |

greatest probability by far, followed by 14, 1110 (WiP, body, feedback and realistic). In fact, the posterior probability that $p_6$ is greater simultaneously than all of the other configuration probabilities is 1.

In addition to deriving the posterior probabilities for the $p_j$ we can find the marginal distributions over each factor separately. For example, the posterior probability for the factor level corresponding to WiP is the sum of the $p_j$ over all configurations corresponding to WiP (i.e. the first element in the configuration is 1). These are shown in table 3 and figure 5. From the means of the distributions in table 3 we can see that considered individually the highest probability factor level corresponds to realistic rendering, followed by teleportation, body and social feedback, with cartoon rendering the lowest. Considering the credible intervals, there is considerable overlap between body and feedback. WiP has probability less than those two and cartoon clearly has the lowest probability. This is also clear from figure 5.

## 3.3. Post-experience questionnaire

The results of the post-experience questionnaire are shown in figure 6. The results conform to the choices made during the VR exposure. The participants were also asked 'In general, do you think that your

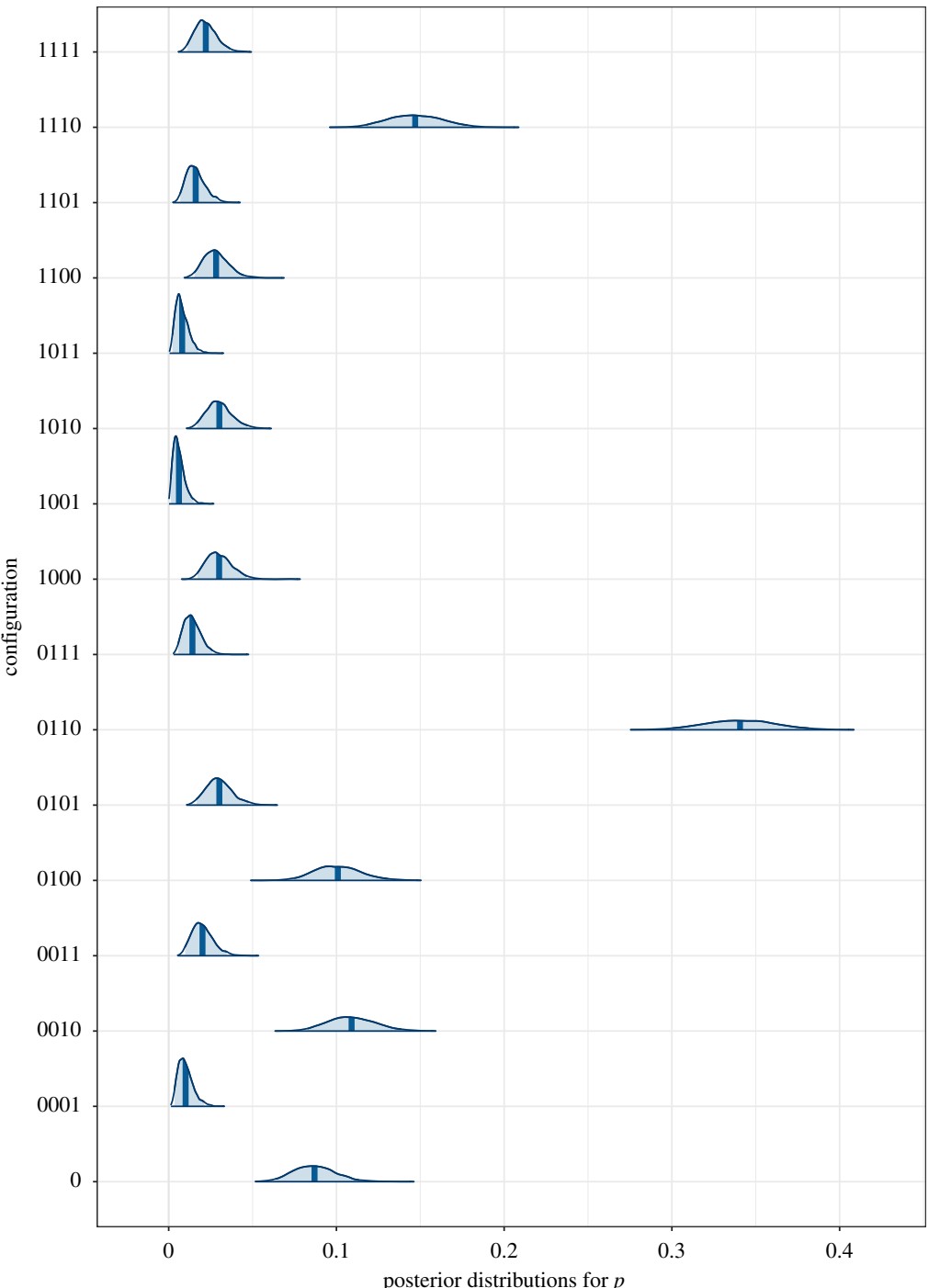

**Figure 4.** Posterior distributions of the $p_j = 0$, $j = 0, 1, \ldots, 15$. The configurations correspond to table 1, where configuration *abcd* corresponds to $a = 0$ (teleportation), $a = 1$ (WiP); $b = 0$ (hands), $b = 1$ (body); $c = 0$ (ignored), $c = 1$ (feedback); $d = 0$ (realistic), $d = 1$ (cartoon). The shaded areas are the 95% credible intervals and the vertical lines are the means of the distributions.

choices within the experiment are consistent with the preferences you have indicated in this questionnaire?' with possible yes/no answers. All 20 participants answered 'yes'.

Participants were also asked to comment about their reasons for their questionnaire responses. With respect to the choice between teletransportation or WiP the most common reason for choosing teletransportation was that it was easier and more comfortable than WiP. The common reason for choosing the full body rather than only hands was that the body resulted in greater realism. Regarding the responsiveness or not of the crowd, the responsive crowd was preferred because this was considered to be more realistic. Again realism was given for the choice of more realistic compared to cartoon rendering.

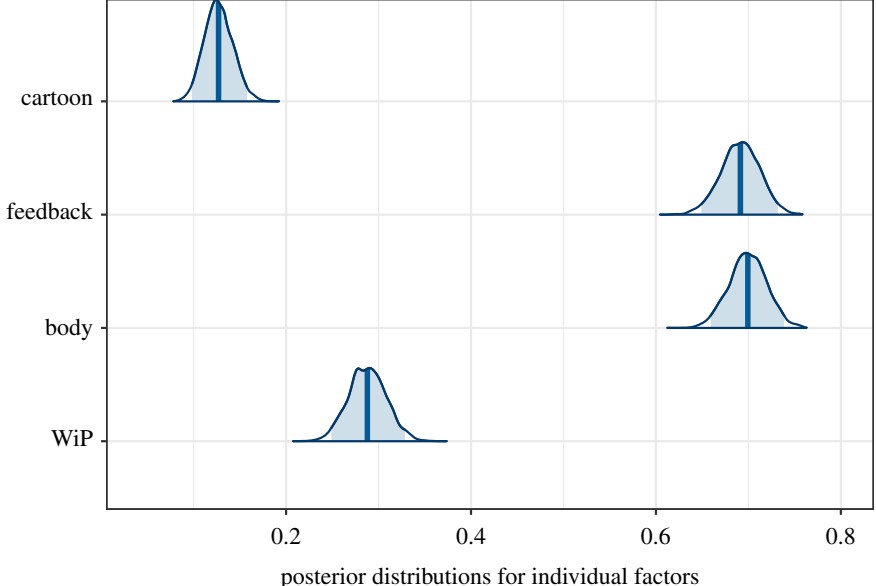

**Figure 5.** Posterior distributions of the marginals for the factor levels. The shaded areas are the 95% credible intervals and the vertical lines are the means of the distributions.

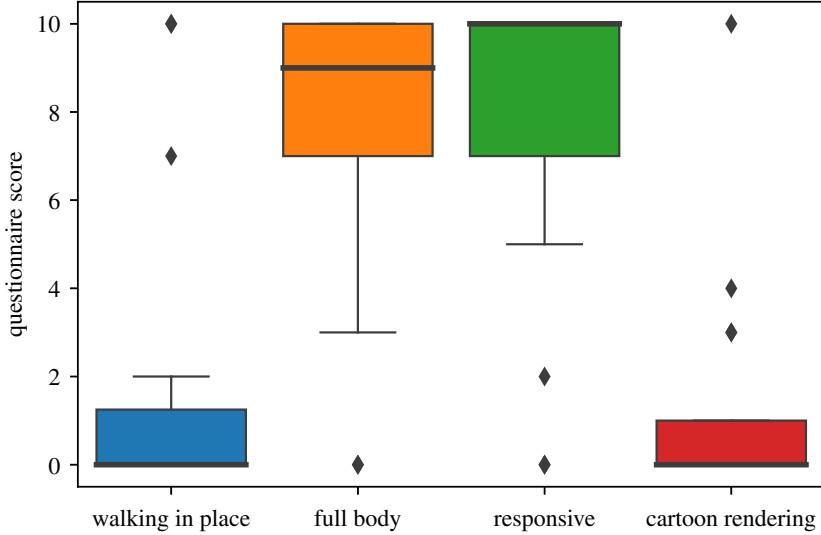

**Figure 6.** Box plots of the results of the post-experience questionnaire. The questions are shown in table 2. The thick horizontal lines are the medians, the boxes are interquartile ranges (IQR), the whiskers range from max(min value, lower quartile $- 1.5 \times$ IQR) to min(max value, upper quartile $+ 1.5 \times$ IQR). Values outside this range are shown as individual points.

### 3.4. Sample size

The results above are based on a sample size of 20 participants. Here, we consider the sensitivity of the results to this sample size. We considered what might have happened had the sample size been 15. We selected 15 of the 20 at random 10 000 times and recomputed the RLscores in each case, finally obtaining a 10 000 × 16 matrix, where each column was a particular configuration and the rows were the results of the 10 000 samples. For each row, we found the minimum RLscore value, and the corresponding configuration. In each case, the minimum RLscore corresponded to the configuration 6 in table 1 (teleportation, body, feedback and realistic).

In addition, in the electronic supplementary material, table S2, we show the configurations sorted by RL score when we consider half of the configuration visits for all 20 subjects. Four of the five top configurations in table 2 also appear in the top 5.

# 4. Discussion

We have introduced a new method for assessing which factors contribute to participants' acceptance of a VR scenario. A RL agent offers choices for changing a factor level periodically during the experience and learns over time which changes are likely to be rejected or accepted by the participant. Eventually, the RL converges on a policy, indicated by the participant accepting no further changes, having reached their optimal configuration. These optimal configurations were consistent across 20 participants.

In contrast with the vast majority of studies of how different factors influence the sense of presence (whether PI or Psi) in virtual environments, we have used simply participant preference, following [35]. The advantage of this is that the results are solely dependent on participant actions during the course of their experience, rather than *post hoc* evaluations based on a questionnaire that may raise issues (such as their sense of 'being there') which might not have occurred to participants at all during their exposure. In this way, we avoid imposing our conceptual framework on participants. Hence, the method allows the assessment of participant preferences without needing post-experimental questionnaires. It is known that questionnaires alone are not satisfactory as these can lead participants to reinterpret the experience in terms consistent with the framing provided by the questionnaire (see [26]). Here, we require participants to make binary choices within the experiment, but this does not frame their choice in any other way than their experience and preferences during the time of the exposure itself. In this particular study, we only used questionnaires as a way to test for consistency between configurations during the experience and final judgements.

The disadvantage is that we do not know the reasons for participant preferences, only the combination of factor levels that they preferred. Although this is beyond the purpose of this particular study, we can gain some insight into how this might be possible through post-exposure interviews. We found that participants tended to quote 'realism' and also 'comfort' to justify their choices. We propose that subsequent use of this method should employ a systematic post-exposure interview, where participants would be asked to explain the reasons for their choices, and then sentiment analysis [51,52] used to rigorously code their responses as in [38].

The example we employed included four binary factors, and the RL method converged, moreover with results that are mainly consistent with previous findings. With respect to the body representation, it has been shown several times that having a body results in higher levels of presence than not having one—for example [27,53]. The illusion of body ownership that can result from having a virtual body has been shown to have a profound influence on various aspects of participant responses, even including the response to pain [54]. Realistic rendering is more likely to result in greater levels of Psi [27,40]. Social feedback is critical in the maintenance of Psi [55]. The anomaly is navigation, where previous research has shown that WiP leads to greater presence than point-and-click methods [39,44]. In interviews with participants after their experience, they reported that WiP resulted in cybersickness most especially when they had to go down the steps in the theatre towards the stage. Hence, participants were more likely to prefer teleportation than WiP. It was considered the easier and more comfortable method to use. However, the particular factors used in this study are not important to the method itself. Any set of factors could have been employed, and the ones we chose were of interest to us in the context of the wider project of which this study is a part.

Regarding experimental configurations, both the visit counts and the scores derived from the Q-table in the RL algorithm accurately reflect participants' questionnaire-based preferences: the five options most preferred by the participants in the confirmation phase were shown by these two measures, as well as their order in ranking, demonstrating internal consistency.

An advantage of the method is that, unlike previous work [27,32,33], the set-up does not imply an *a priori* ordering among the factor levels (e.g. it does not start out with the assumption that having a body is preferable to having only hands). The method proposes options among which the participant can choose, without any implication that one is superior to another—they are just different. This has the benefit of allowing participants to select between levels of an experimental factor which have no implied ordering, for example, navigation methods that are radically different from one another. This is in contrast with [27] where participants were required to select the level of a factor in order, with the implication that each successive level was superior to the previous one.

It would be straightforward to extend the proposed method to continuous variables, as long as the continuous variables considered can be converted to an array of discrete samples. This would only require adapting the interface panels (figure 2) in such a way as to propose choices among the

sampled values, or replace the RL method used, Q-learning, with one adapted to continuous states, such as gradient-based methods [47].

Finally, an advantage is that the method obtains its result through a probabilistic search through the space of factor levels, rather than requiring a $2^4$ factorial design. Hence the method would be extendable to more than four factors, although the relationship between the number of factor levels and sample size needed to obtain a convergent and consistent result needs to be explored. In addition, although this particular study was limited to binary factors, there is no special reason for this limitation. With multiple levels per factor, however, the RL would almost certainly require more time to converge because the space of exploration would be much greater. This remains a question for further work. Moreover, any method such as this cannot guarantee a 'global maximum' only that a local maximum has been reached. However, this is mitigated by the fact of having many participants engage in the procedure, each starting from a different initial configuration.

We have addressed some limitations of the study above. Each individual made choices according to their preference, but there is an overall consistency between the individual results. However, we do not have much insight into the reasons for their choices. This method has to be combined with post-exposure interviews, but where suggestions are not put into the minds of participants through a questionnaire, rather where interviewer and interviewee can together explore the reasons for their choices. It might be very interesting to discover whether the view of researchers that presence is the primary illusion in virtual environments responsible for the overall quality of experience of participants is actually supported by participants themselves. If a questionnaire specifically asks about presence then participants will give a *post hoc* answer irrespective of how important this had actually been. However, if participants are asked why they made a particular choice, the answer might be quite different. In our study, participants often quoted 'realism' as the reason for their choices. However, this is still rather vague, and a post-exposure interview would help to throw light on this. Second, we have suggested that choices might be on continuous scales and be multi-level rather than binary, and that more than four factors could be accommodated. While this is the case, at this time we have no evidence about the practical implications. How long would convergence take (if any) for each participant and how many participants would be needed? How many participants would be needed to validate inter-subject consistency? These are questions for subsequent research into this method.

To study this method in the future, we wish to consider alternate measures derived from the task. In particular, we would like to explore whether time-based measures, i.e. how long participants stayed in the configurations, can reliably predict participant preferences. The reliability of visit counts already points towards this direction. It is possible that the time participants spent in a given configuration, even taking into account the time spent exploring the different choices, is a more reliable indicator than the visit counts. Time measures become particularly relevant in the context of deep RL methods, as no explicit value-table is available to compute easily a RL score such as the one used in this method.

A further refinement could also take into account the cost of the various factor levels. For example, the probability of an option being displayed could be set initially to be inversely proportional to its cost (e.g. the cost of programming or maintenance or the hardware requirement for efficient implementation). In this way, if such a factor level actually rises to the forefront and becomes part of the RL policy, this would be a strong indication of its importance and therefore may be worth the cost.

Finally, we would like to explore to what extent the method introduced can be used to assess participant preferences using *implicit* measures, rather than explicit participant choices. By this we refer to various indicators of participant state such as the interpersonal distances they maintain with virtual characters, gaze or navigation patterns. Combining this method with implicit measures would allow for far richer and more systematic preference evaluation, especially for experiences where the introduction of panels and explicit tasks would not be a viable option. Here, we envisage the RL operating in the background and choosing how to configure aspects of the virtual environment depending on participant behaviour.

Overall we propose that the method introduced here is particularly well adapted to the evaluation of preferences over factors that influence the quality of VR experiences. Given the growing adoption of VR, this method and some of its possible extensions outlined could become the VR equivalent of A/B testing [56], common today in the evaluation of web technologies.

Ethics. The experiment was approved by the Comissió de Bioètica de la Universitat de Barcelona (IRB00003099). Prior to participating in the experiment, participants read and signed an information sheet and a consent form, thus giving written informed consent. Participants were informed both verbally and in writing that they were free to withdraw from the experiment at any time without giving any reasons.

Data accessibility. Data and all code are available through the electronic supplementary material, data S1 where a detailed description is given on how to access the data and execute the corresponding code.

Authors' contributions. J.L. contributed to the design of the experiment, implemented the VR scenario, ran the study with participants, collected and tabulated the data, and contributed towards writing the paper. A.B. contributed to the implementation of the VR scenario and reviewed the paper. R.O. contributed to the implementation of the VR scenario and reviewed the paper. G.S. contributed to the execution of the experiment and reviewed the paper. D.B. contributed to the design of the experiment and reviewed the paper. M.S. formulated the original concept, contributed to the design of the experiment, analysed the results, wrote the draft of the paper and obtained the funding.

Competing interests. The authors declare no competing interests.

Funding. This research is supported by the European Research Council Advanced grant Moments in Time in Immersive Virtual Environments (MoTIVE) grant no. 742989 and all authors were funded by this grant except for G.S. who is supported by 'la Caixa' Foundation (ID 100010434) with Fellowship code no. LCF/BQ/DR19/11740007.

Acknowledgements. We thank Dr Ilias Bergström for supplying the string quartet animation from [31].

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
