## [Peer Review File · Royal Society Open Science]

Review History

RSOS-210537.R0 (Original submission)

Review form: Reviewer 1 (Matthew Smith)

Is the manuscript scientifically sound in its present form?

Yes

Are the interpretations and conclusions justified by the results?

No

Is the language acceptable?

Yes

Do you have any ethical concerns with this paper?

No

Have you any concerns about statistical analyses in this paper?

No

Recommendation?

Major revision is needed (please make suggestions in comments)

Comments to the Author(s)

This manuscript proposes novel methods and processes for determining participant preferences along four dimensions when using virtual reality. The approach is timely and has merit; however there are some issues with the manuscript that need to be addressed before it is ready for publication.

From a big picture perspective, I found it somewhat difficult to make the connection between the method and measurement of quality of experience. My own research is situated at the intersection of learning design and user experience research, so I am interested in how the authors operationalize quality of experience (QoE) within the context of this study specifically and in VR in general; however, I was not able to find a discrete operationalization. Instead, the authors indicate that QoE is most often assessed through measures of presence (PI and Psi) in VR research – which is true. But there is more to user experience in VR than presence. For example, (virtual) context plays a role, as does the users' goal-orientation (which is dependent on the task).

The authors state the proposed method could be used to assess factors contributing to participants' acceptance of the VR scenario (P. 23 l. 8). This was not made clear. Was it acceptance? Or general preference for interaction within a VR space? Acceptance was not measured. P. 25 line 23 they state their approach is well adapted to evaluate preferences over "factors that influence the quality of VR experiences." This needs to be clarified.

The paper is methodological in nature. That is, it seeks to streamline methods of preference assessment. This is a strength. However, how this intersects with the conceptual framing presented in the article introduction is unclear and needs to be further developed. I question the extent to which this approach is connected with Psi and PI, as opposed to being a manifestation of user preferences. The authors do not explicitly claim that there is a connection, but given the focus on presence in the introduction and the four factors selected for assessing preferences, an implicit connection is strongly suggested. There seems to be a general conflation of quality of experience with presence.

The description of the RL design was clear, assumptions were reasonable, and tradeoffs made sense. However, the value of the RL design is not clear. How is this approach superior to asking user preferences randomly or in a step-wise manner? Does the RL approach introduce bias? The RL predicts what the user will prefer, which could promote efficiency. But the findings presented in the paper could very well have been arrived at without the RL agent. Why is it necessary?

The authors suggest that their approach could be modified to accommodate continuous variables. This is a very interesting suggestion and should be considered as a potential direction for future research. However, it also highlights an issue with the current study that is unsettling – the binary nature of the variables and the resultant options presented to users. It appears the authors sought maximum variation relative to factors influencing sense of presence (again, suggesting an implicit connection between the four factors and presence). But this approach may have some pitfalls. For example, participants' preference for navigation methods that have less likelihood for cybersickness seems rather obvious. If WIP caused adverse effects, then, behaviorally speaking, the logical response would be to choose the more reinforcing (and less punishing) method. Does this influence quality of experience? Yes. Does it influence perceptions of PI and Psi? Potentially, but the question is whether this is the primary impetus for user preference. From the perspective of affordances, walking in place vs. point-and-click are fundamentally different both qualitatively and quantitatively, with one requiring physical effort and the other not – would preferences

change if for example joystick navigation were used instead of walking-in-place? The cel-shading method used was somewhat cartoonish, but ultimately used the same avatars and the same environments. The rendering was still fairly realistic. Was the juxtaposition sufficient? Would you see different results with lower polygon environments or less sophisticated lighting, shadows, etc? Finally, the authors state in abstract: "Questionnaires are problematic since the questions can frame how participants think about their experience and cannot easily take account of non-additivity amongst the various factors." Doesn't the approach here frame how participants think about their experience as well?

Ultimately, the authors are encouraged to better differentiate quality of experience from presence, to better operationalize quality of experience, and to make explicit connections between factors influencing quality of experience and the construct itself. In the discussion, the implications of this could be discussed in relation to the broader construct of presence. Further, greater attention should be given to the limitations of this study in the discussion section – currently, limitations are absent.

Below are more minor issues for the authors to attend to:

P.4 l. 19 "respond realistically to events and situations" – please clarify further. E.g., "respond realistically to the events and situations presented in the digital environment."

P.4 l. 24. "from a cognitive viewpoint it makes no sense" – perhaps an overstatement. Perhaps consider "is perplexing"?

P.4. l. 24-29. Incomplete sentence.

P. 4 l. 55-59. Refers to 2 constructs, then states this is the target sensation. Which one? Unclear.

P. 4 l. 60. "they were in a virtual environment with all factors at a low level" – juxtaposed with highest levels in line 53. Unclear how factors were at a low/high level. Please provide further detail on the notion of levels in this context. An example would likely be sufficient.

P. 6. l. 9. "decision only consists on" – did authors mean "depends on"?

P. 13 l. 30: "Doing this at this moment also increased the time that they would be without wearing the HMD." – This is awkward.

P. 13 l. 54: did the authors mean "realistic"?

P. 23 l. 54-60. Authors state "The method proposes options amongst which the participant can choose, without prioritizing one over another. This has the benefit of allowing the testing of experimental factors which do not have a clear hierarchical structure such as, for example, navigation methods that are radically different from one another." However, doesn't the RL introduce an individualized prioritization? It's unclear if this is what you meant.

Minor grammatical issues were noted throughout, as were issues with citation style. These will likely be attended to during copy editing, but worth bringing to the authors' attention.

Review form: Reviewer 2

Is the manuscript scientifically sound in its present form?

Yes

Are the interpretations and conclusions justified by the results?

Yes

Is the language acceptable?

Yes

Do you have any ethical concerns with this paper?

No

Have you any concerns about statistical analyses in this paper?

No

Recommendation?

Accept with minor revision (please list in comments)

Comments to the Author(s)

The study investigated the participants' reaction and preferences of their control mechanism and appearance in VR. The paper is well-written and it is pleasant to read such an interesting approach in VR. Here are only a few minor points for the authors to consider during the revision:

- The motivation of the application of the RL was not noticeably clear for me. At first it reads like the authors use RL as a "recommendation" system to try to make the participants accept their own preferred consideration, which is potentially problematic since it would make the experiment exposure thus the independent variable of each participant unique. Then later on the RL score was used as an evaluation metric for ranking the preference of the configurations. This confuses me since if RL was used as part of the experiment, how could the value that the algorithm generated be used as a dependent variable, considering that the way in which the parameters of the algorithm was tuned would largely influence the RL score?
- Page 6 line 24 to 60 are the details of the experiment should be placed into the method part. Instead, I suggest the authors should briefly describe the design of the experiment and the motivations of using RL in the end of the introduction.
- The four experimental factors are the most important part of the study but there was no illustration of them. I suggest adding a figure clearly showing the four factors.
- Figure 3. I am not sure how informative this plot is. If the author wants to keep it, maybe consider replacing the Spanish with English to increase the readability of the figure.
- Page 18. Table without caption.
- Page 24. The authors used 3 paragraphs to picture the future work that they want to conduct. I suggest replacing them with some limitations of the study.
- Since this paper is likely of interest to a wide multi-disciplinary community (both ML and VR researchers), the authors should consider positioning the work to also be accessible to machine learning researchers. While the paper relies on a straightforward RL technique (Q-learning) and does not make a machine learning contribution, the proposed task definition may be of interest to the wider ML community. Positioning this task in relation to well-studied RL tasks would be beneficial.

Decision letter (RSOS-210537.R0)

Dear Dr Slater,

The Editors assigned to your paper RSOS-210537 "Evaluating Participant Responses to a Virtual Reality Experience Using Reinforcement Learning" have now received comments from reviewers and would like you to revise the paper in accordance with the reviewer comments and any comments from the Editors. Please note this decision does not guarantee eventual acceptance.

Please submit your revised manuscript and required files (see below) no later than 21 days from today's (ie 19-Jul-2021) date. Note: the ScholarOne system will 'lock' if submission of the revision is attempted 21 or more days after the deadline. If you do not think you will be able to meet this deadline please contact the editorial office immediately.

on behalf of Dr Mirco Musolesi (Associate Editor) and Marta Kwiatkowska (Subject Editor)
openscience@royalsociety.org

Associate Editor Comments to Author (Dr Mirco Musolesi):

The reviewer found the paper interesting and insightful, but they also raised a series of concerns that should be addressed in a revised version of the manuscript. Two points that are quite important to address are the operationalization of the proposed method and the limitations/applicability.

Reviewer comments to Author:

Reviewer: 1

Comments to the Author(s)

This manuscript proposes novel methods and processes for determining participant preferences along four dimensions when using virtual reality. The approach is timely and has merit; however

there are some issues with the manuscript that need to be addressed before it is ready for publication.

From a big picture perspective, I found it somewhat difficult to make the connection between the method and measurement of quality of experience. My own research is situated at the intersection of learning design and user experience research, so I am interested in how the authors operationalize quality of experience (QoE) within the context of this study specifically and in VR in general; however, I was not able to find a discrete operationalization. Instead, the authors indicate that QoE is most often assessed through measures of presence (PI and Psi) in VR research – which is true. But there is more to user experience in VR than presence. For example, (virtual) context plays a role, as does the users' goal-orientation (which is dependent on the task).

The authors state the proposed method could be used to assess factors contributing to participants' acceptance of the VR scenario (P. 23 l. 8). This was not made clear. Was it acceptance? Or general preference for interaction within a VR space? Acceptance was not measured. P. 25 line 23 they state their approach is well adapted to evaluate preferences over "factors that influence the quality of VR experiences." This needs to be clarified.

The paper is methodological in nature. That is, it seeks to streamline methods of preference assessment. This is a strength. However, how this intersects with the conceptual framing presented in the article introduction is unclear and needs to be further developed. I question the extent to which this approach is connected with Psi and PI, as opposed to being a manifestation of user preferences. The authors do not explicitly claim that there is a connection, but given the focus on presence in the introduction and the four factors selected for assessing preferences, an implicit connection is strongly suggested. There seems to be a general conflation of quality of experience with presence.

The description of the RL design was clear, assumptions were reasonable, and tradeoffs made sense. However, the value of the RL design is not clear. How is this approach superior to asking user preferences randomly or in a step-wise manner? Does the RL approach introduce bias? The RL predicts what the user will prefer, which could promote efficiency. But the findings presented in the paper could very well have been arrived at without the RL agent. Why is it necessary?

The authors suggest that their approach could be modified to accommodate continuous variables. This is a very interesting suggestion and should be considered as a potential direction for future research. However, it also highlights an issue with the current study that is unsettling – the binary nature of the variables and the resultant options presented to users. It appears the authors sought maximum variation relative to factors influencing sense of presence (again, suggesting an implicit connection between the four factors and presence). But this approach may have some pitfalls. For example, participants' preference for navigation methods that have less likelihood for cybersickness seems rather obvious. If WIP caused adverse effects, then, behaviorally speaking, the logical response would be to choose the more reinforcing (and less punishing) method. Does this influence quality of experience? Yes. Does it influence perceptions of PI and Psi? Potentially, but the question is whether this is the primary impetus for user preference. From the perspective of affordances, walking in place vs. point-and-click are fundamentally different both qualitatively and quantitatively, with one requiring physical effort and the other not – would preferences change if for example joystick navigation were used instead of walking-in-place? The cel-shading method used was somewhat cartoonish, but ultimately used the same avatars and the same environments. The rendering was still fairly realistic. Was the juxtaposition sufficient? Would you see different results with lower polygon environments or less sophisticated lighting, shadows, etc? Finally, the authors state in abstract: "Questionnaires are problematic since the questions can frame how participants think about their experience and cannot easily take account

of non-additivity amongst the various factors.” Doesn’t the approach here frame how participants think about their experience as well?

Ultimately, the authors are encouraged to better differentiate quality of experience from presence, to better operationalize quality of experience, and to make explicit connections between factors influencing quality of experience and the construct itself. In the discussion, the implications of this could be discussed in relation to the broader construct of presence. Further, greater attention should be given to the limitations of this study in the discussion section – currently, limitations are absent.

Below are more minor issues for the authors to attend to:

P.4 l. 19 “respond realistically to events and situations” – please clarify further. E.g., “respond realistically to the events and situations presented in the digital environment.”

P.4 l. 24. “from a cognitive viewpoint it makes no sense” – perhaps an overstatement. Perhaps consider “is perplexing”?

P.4. l. 24-29. Incomplete sentence.

P. 4 l. 55-59. Refers to 2 constructs, then states this is the target sensation. Which one? Unclear.

P. 4 l. 60. “they were in a virtual environment with all factors at a low level” – juxtaposed with highest levels in line 53. Unclear how factors were at a low/high level. Please provide further detail on the notion of levels in this context. An example would likely be sufficient.

P. 6. l. 9. “decision only consists on” – did authors mean “depends on”?

P. 13 l. 30: “Doing this at this moment also increased the time that they would be without wearing the HMD.” – This is awkward.

P. 13 l. 54: did the authors mean “realistic”?

P. 23 l. 54-60. Authors state “The method proposes options amongst which the participant can choose, without prioritizing one over another. This has the benefit of allowing the testing of experimental factors which do not have a clear hierarchical structure such as, for example, navigation methods that are radically different from one another.” However, doesn’t the RL introduce an individualized prioritization? It’s unclear if this is what you meant.

Minor grammatical issues were noted throughout, as were issues with citation style. These will likely be attended to during copy editing, but worth bringing to the authors’ attention.

Reviewer: 2

Comments to the Author(s)

The study investigated the participants’ reaction and preferences of their control mechanism and appearance in VR. The paper is well-written and it is pleasant to read such an interesting approach in VR. Here are only a few minor points for the authors to consider during the revision:

- The motivation of the application of the RL was not noticeably clear for me. At first it reads like the authors use RL as a “recommendation” system to try to make the participants accept their own preferred consideration, which is potentially problematic since it would make the experiment exposure thus the independent variable of each participant unique. Then later on the RL score was used as an evaluation metric for ranking the preference of the configurations. This

confuses me since if RL was used as part of the experiment, how could the value that the algorithm generated be used as a dependent variable, considering that the way in which the parameters of the algorithm was tuned would largely influence the RL score?

- Page 6 line 24 to 60 are the details of the experiment should be placed into the method part. Instead, I suggest the authors should briefly describe the design of the experiment and the motivations of using RL in the end of the introduction.
- The four experimental factors are the most important part of the study but there was no illustration of them. I suggest adding a figure clearly showing the four factors.
- Figure 3. I am not sure how informative this plot is. If the author wants to keep it, maybe consider replacing the Spanish with English to increase the readability of the figure.
- Page 18. Table without caption.
- Page 24. The authors used 3 paragraphs to picture the future work that they want to conduct. I suggest replacing them with some limitations of the study.
- Since this paper is likely of interest to a wide multi-disciplinary community (both ML and VR researchers), the authors should consider positioning the work to also be accessible to machine learning researchers. While the paper relies on a straightforward RL technique (Q-learning) and does not make a machine learning contribution, the proposed task definition may be of interest to the wider ML community. Positioning this task in relation to well-studied RL tasks would be beneficial.

===PREPARING YOUR MANUSCRIPT===

===PREPARING YOUR REVISION IN SCHOLARONE===

Author's Response to Decision Letter for (RSOS-210537.R0)

See Appendix A.

Decision letter (RSOS-210537.R1)

Dear Dr Slater,

It is a pleasure to accept your manuscript entitled "Evaluating Participant Responses to a Virtual Reality Experience Using Reinforcement Learning" in its current form for publication in Royal Society Open Science. The comments of the reviewer(s) who reviewed your manuscript are included at the foot of this letter.

on behalf of Dr Mirco Musolesi (Associate Editor) and Marta Kwiatkowska (Subject Editor)
openscience@royalsociety.org

Associate Editor Comments to Author (Dr Mirco Musolesi):
Associate Editor

Comments to the Author:

The authors carefully addressed the issues raised by the reviewers. The revised version of the manuscript is substantially improved. I would recommend this manuscript for acceptance.

Appendix A

RSOS-210537 "Evaluating Participant Responses to a Virtual Reality Experience Using Reinforcement Learning"

We thank the editor and reviewers for helpful comments and suggestions. Below the comments are in blue and our responses in black. We have uploaded two versions of the paper, without tracked changes (the main manuscript) and with tracked changes. The one with tracked changes has line numbers and these are referred to in our responses below.

Reviewer: 1

Comments to the Author(s)

This manuscript proposes novel methods and processes for determining participant preferences along four dimensions when using virtual reality. The approach is timely and has merit; however there are some issues with the manuscript that need to be addressed before it is ready for publication.

We thank the reviewer for helpful comments which we address point by point.

From a big picture perspective, I found it somewhat difficult to make the connection between the method and measurement of quality of experience. My own research is situated at the intersection of learning design and user experience research, so I am interested in how the authors operationalize quality of experience (QoE) within the context of this study specifically and in VR in general; however, I was not able to find a discrete operationalization. Instead, the authors indicate that QoE is most often assessed through measures of presence (PI and Psi) in VR research—which is true. But there is more to user experience in VR than presence. For example, (virtual) context plays a role, as does the users' goal-orientation (which is dependent on the task).

In this research we have taken a straightforward view of participant experience - it is just a matter of their preference. The discussion on presence earlier in the paper was just to introduce the normal approach that is considered in virtual reality as a way of assessing user experience. Our approach is not based on presence at all. The participant is offered a series of binary choices (between two levels of a factor) and chooses based solely on their preference, with no other hint given to them about the criteria on which to base their choice. This the same approach as in 'Evaluating Virtual Reality Experiences Through Participant Choices' (Murcia-López et al., 2020), <https://ieeexplore.ieee.org/abstract/document/9089592> with the fundamental difference that the sequence of options offered to the participants is based on a reinforcement learning algorithm, that eventually converges so that no further changes were accepted by the participant, and with a high degree of similarity across participants. In the revised paper, we have made our overall approach clearer and emphasised that there was not an attempt to measure presence in this experiment, but only participant preference amongst the factors [page 5, lines 7-9, 18-20, 25-29; page 13, lines 4-6, 28-29; page 23, lines 9-30; page 25, lines 8-26].

The authors state the proposed method could be used to assess factors contributing to participants' acceptance of the VR scenario (P. 23 l. 8). This was not made clear. Was it acceptance? Or general preference for interaction within a VR space? Acceptance was not measured. P. 25 line 23 they state their approach is well adapted to evaluate preferences over "factors that influence the quality of VR experiences." This needs to be clarified.

Acceptance or not is just observed - it is not measured but is the action that participants take. Participants were offered choices to change a configuration (for example, seeing the world rendered in cartoon or more realistic rendering), and accepted a change to the current configuration or not. Their only criterion for acceptance of the change was that they preferred it. Eventually a convergence took place where no further suggestions of change were accepted, the implication being that for the participant an optimal configuration would have been reached. This is discussed in several places throughout the paper, but in particular we have added page 23, lines 9-22.

The paper is methodological in nature. That is, it seeks to streamline methods of preference assessment. This is a strength. However, how this intersects with the conceptual framing presented in the article introduction is unclear and needs to be further developed. I question the extent to which this approach is connected with Psi and PI, as opposed to being a manifestation of user preferences. The authors do not explicitly claim that there is a connection, but given the focus on presence in the introduction and the four factors selected for assessing preferences, an implicit connection is strongly suggested. There seems to be a general conflation of quality of experience with presence.

We have made it clear in the revised version that this method is nothing *per se* to do with presence. It is based solely on participant preference. An earlier similar method (without the Reinforcement Learning aspect) was used to assess presence, but this is not the case here. We have mentioned this on page 3 line 23 – page 4, line 9.

The description of the RL design was clear, assumptions were reasonable, and tradeoffs made sense. However, the value of the RL design is not clear. How is this approach superior to asking user preferences randomly or in a step-wise manner? Does the RL approach introduce bias? The RL predicts what the user will prefer, which could promote efficiency. But the findings presented in the paper could very well have been arrived at without the RL agent. Why is it necessary?

It would be possible to offer, at random, choices to participants. In fact the RL method starts out that way. But the value of the RL method is that it eventually converges to the extent that participants have reached an optimal configuration such that they do not want to accept further changes. Up to the problem that only a 'local maximum' might have been reached (which is already mentioned in the paper) we can think of the RL finding such an optimal solution. If changes were presented at random it is not clear what the stopping rule would be. One would end up with a sequence of randomly accepted/rejected changes that would not have any particular structure. One would have to somehow keep a record of, for example, how many times a suggested change had been accepted or rejected, and then try to make sense of that, and in the end probably invent something like RL in order to give structure to the changes. An alternative is for participants to choose which changes to make by themselves, which has been used in previous methods (see Skarbez et al. https://discovery.ucl.ac.uk/id/eprint/1550432/1/Slater_SkarbezTVCG.pdf). This is fine but the RL method explores the space in a systematic way, and does not rely on memory of the participants. This is not so important in this particular experimental setup but it could be very important when there are more than 4 factors. This has been explained on page 24, lines 9-11; page 25, lines 21-22].

Regarding the possible introduction of bias, it is important to consider that the RL method does not use the estimated values across participants, but rather it reinitializes the value matrix for each participant. This implies that there are no assumptions regarding whether the

preferences of one participant are consistent with the preferences of other participants. Each participant is treated separately [page 16, lines 14-15].

If we do not want to ask the participant to remember all possible choices, the traditional way of arriving at the same findings would be to do a factorial design, which would significantly increase the time and resources needed.

The authors suggest that their approach could be modified to accommodate continuous variables. This is a very interesting suggestion and should be considered as a potential direction for future research. However, it also highlights an issue with the current study that is unsettling—the binary nature of the variables and the resultant options presented to users. It appears the authors sought maximum variation relative to factors influencing sense of presence (again, suggesting an implicit connection between the four factors and presence).

Our contribution is essentially methodological, and binary choices were sufficient to demonstrate the benefits of this approach, when compared to more typical experimental designs, such as a factorial design. It should be noted that we did not at all consider presence, except in the introduction to the paper. Participants worked solely on the basis of their preference. Presence was never mentioned to them. However, factors that we chose for the manipulation were indeed motivated by previous research on presence (as well as our interest in these factors as part of a wider project) and clearly presence is an important aspect of the virtual reality experience. But this is a quite separate issue from the instructions given to participants, which did not mention presence, nor can we infer that presence was implicated in their choices. This has been made clear on page 5, lines 18-20, 25-29; page 13, lines 4-6, 28-29; page 23, lines 9-14.

But this approach may have some pitfalls. For example, participants' preference for navigation methods that have less likelihood for cybersickness seems rather obvious.

This was a particular idiosyncrasy of this particular experimental scenario. We explained this in the paper only to draw attention to the choices made that were unexpected compared to previous findings in the literature (i.e., that walking-in-place is typically preferred to point-and-click when presence is the goal). This is independent of the method as a whole, and as the reviewer has noted, this is primarily a methodological paper, so the particular factors we chose to study should be considered as illustrative of the method only. We have made this clear on page 24 lines 9-11.

If WIP caused adverse effects, then, behaviorally speaking, the logical response would be to choose the more reinforcing (and less punishing) method. Does this influence quality of experience? Yes. Does it influence perceptions of PI and Psi? Potentially, but the question is whether this is the primary impetus for user preference.

We have not argued that presence was involved in participant choices nor that it was the primary impetus. The only instruction to participants was to choose on the basis of their personal preference. We have now pointed out that *we do not wish to impose our notion of quality of experience as presence on the participants*, hence their choices were motivated solely by preference (for whatever reason - in fact the reason is irrelevant to the method). Page 23, lines 9-22.

It should also be noted that the preference for teleport compared to WIP was not universal, and at least one participant did prefer WIP. This is consistent with the fact that they chose

what they preferred, and not what was most comfortable, or what maximized their feeling of presence.

From the perspective of affordances, walking in place vs. point-and-click are fundamentally different both qualitatively and quantitatively, with one requiring physical effort and the other not—would preferences change if for example joystick navigation were used instead of walking-in-place?

We have made it clear in the new version that the issue of walking-in-place compared to point-and-click is a contingent aspect of this particular experimental setup and has nothing to do with the method in itself, page 24, lines 9-11.

The cel-shading method used was somewhat cartoonish, but ultimately used the same avatars and the same environments. The rendering was still fairly realistic. Was the juxtaposition sufficient? Would you see different results with lower polygon environments or less sophisticated lighting, shadows, etc?

This is an important issue. Of course we cannot answer this based on these data. However, the same setup with RL could be used for any set of factors including those suggested.

Finally, the authors state in abstract: “Questionnaires are problematic since the questions can frame how participants think about their experience and cannot easily take account of non-additivity amongst the various factors.” Doesn’t the approach here frame how participants think about their experience as well?

We do not think that these can be compared in the same way. A questionnaire frames how participants think about their experience through focussing their attention on particular aspects that may not have figured in their experience at all. For example, we can ask participants about their “sense of being there”, but this may not have been important in their experience, but they will nevertheless give an answer to a question. In this new proposed method everything is based directly on immediate experience - they experience, for example, the characters around looking at them or ignoring them. They choose that which they prefer. The method does not suggest anything that is actually outside of the direct experience of the participants.

We also point to the fact that the task of reporting preferences was presented in such a way that the options available were characterized with visual icons, avoiding verbal language, again to minimize possible biases introduced through the wording of the questions.

We have made this distinction clear on page 23 lines 9-22.

Ultimately, the authors are encouraged to better differentiate quality of experience from presence, to better operationalize quality of experience, and to make explicit connections between factors influencing quality of experience and the construct itself. In the discussion, the implications of this could be discussed in relation to the broader construct of presence. Further, greater attention should be given to the limitations of this study in the discussion section—currently, limitations are absent.

We now make it clear that although this type of approach that we propose can (and in a different version without the RL, has) been used for the assessment of presence, this is not our intention here. We do not further amplify the construct of presence, since we are not

dealing with presence in this study. However, we have amplified the limitations of the study, page 23 lines 23-30; page 25, lines 8-26.

Below are more minor issues for the authors to attend to:

P.4 I. 19 “respond realistically to events and situations” – please clarify further. E.g., “respond realistically to the events and situations presented in the digital environment.”

We have changed to “... in the virtual environment.”

P.4 I. 24. “from a cognitive viewpoint it makes no sense” — perhaps an overstatement. Perhaps consider “is perplexing”?

This phrase is overstated. We have simplified to “...when talking to a virtual audience, people with fear of public speaking should display anxiety, and this does happen – e.g. [19, 20] – even though all participants know that there is no actual audience there.”

P.4. I. 24-29. Incomplete sentence.

P. 4 I. 55-59. Refers to 2 constructs, then states this is the target sensation. Which one? Unclear.

We have now made it clear that in that paper 50% of the participants were asked to pay attention to their sensation of ‘place illusion’ and the other 50% to their sense of ‘plausibility’. There were therefore two different targets, depending on the group to which participants had been assigned, page 3, line 34 – page 4, line 3.

P. 4 I. 60. “they were in a virtual environment with all factors at a low level”— juxtaposed with highest levels in line 53. Unclear how factors were at a low/high level. Please provide further detail on the notion of levels in this context. An example would likely be sufficient.

This has now been explained with examples in the same paragraph.

P. 6. I. 9. “decision only consists on” – did authors mean “depends on”?

The phrase has been simplified to: “...but here they only have to accept or reject a change proposed by the RL agent”.

P. 13 I. 30: “Doing this at this moment also increased the time that they would be without wearing the HMD.” — This is awkward.

The sentence is redundant and has been deleted.

P. 13 I. 54: did the authors mean “realistic”?

I think that this referred to the word ‘reactive’ in the Figure 3 caption (B). However, we have replaced this figure with another so that this issue no longer applies.

P. 23 I. 54-60. Authors state “The method proposes options amongst which the participant can choose, without prioritizing one over another. This has the benefit of allowing the testing of experimental factors which do not have a clear hierarchical structure such as, for

example, navigation methods that are radically different from one another.” However, doesn't the RL introduce an individualized prioritization? It's unclear if this is what you meant.

Our choice of the word 'hierarchical' was not correct. There are cases when it might be thought that there is a natural ordering in levels of a factor: for example, if the factor were 'field of view' then it might be thought that a wider field of view would be preferred over a narrow field of view (depending on the application). Or, for example, that 'really walking' through an environment might be preferred to 'point-and-click' for navigation. However, the method we have employed does not require or imply any ordering. For example, there could be several different methods of navigation being tested where there is no intrinsic ordering amongst them. This has now been expanded on page 24 lines 17-25.

Minor grammatical issues were noted throughout, as were issues with citation style. These will likely be attended to during copy editing, but worth bringing to the authors' attention.

Reviewer: 2

Comments to the Author(s)

The study investigated the participants' reaction and preferences of their control mechanism and appearance in VR. The paper is well-written and it is pleasant to read such an interesting approach in VR.

We thank the reviewer for these comments and helpful points that serve to improve the paper.

Here are only a few minor points for the authors to consider during the revision:

- The motivation of the application of the RL was not noticeably clear for me. At first it reads like the authors use RL as a “recommendation” system to try to make the participants accept their own preferred consideration, which is potentially problematic since it would make the experiment exposure thus the independent variable of each participant unique. Then later on the RL score was used as an evaluation metric for ranking the preference of the configurations. This confuses me since if RL was used as part of the experiment, how could the value that the algorithm generated be used as a dependent variable, considering that the way in which the parameters of the algorithm was tuned would largely influence the RL score?

The RL system was not used as a recommendation system. Rather, we used the RL method to elicit the preferences of each user. In RL terms: the value of each combination of experimental factors is estimated in each line of the Q-table. Therefore, the Q-table is an estimation of the true preference values, which are unknown.

The RL algorithm tends to balance exploration of the space of possible configurations with convergence towards the configurations that have most value. At the design phase we estimated this could be a good way to evaluate participant preferences.

The fact that the post-experimental questionnaires are consistent with the scores as estimated by the RL method suggests this particular metric is indeed useful to estimate participant preferences.

- Page 6 line 24 to 60 are the details of the experiment should be placed into the method part. Instead, I suggest the authors should briefly describe the design of the experiment and the motivations of using RL in the end of the introduction.

This is always a difficult issue. In those passages we are in the process of motivating the experiment, and describing the scenario in which it takes place. It is part of the introduction in the sense that the rest of the paper wouldn't make sense without it, whereas the strict methods part could even be moved to the end of without seriously compromising understandability. We prefer to keep those passages as part of the introduction unless there is a strong view to move them to the start of the methods section.

- The four experimental factors are the most important part of the study but there was no illustration of them. I suggest adding a figure clearly showing the four factors.

Some of the differences are shown in Figure 1, but the images were rather dark so hard to see. They have been brightened slightly to make the difference between cartoon and realistic rendering clearer. The active or passive characters are mentioned in Figure 1B and D. The navigation method and body or hands are hard to show in a picture, since they are dynamic (e.g., the person has to look towards their body). However, all the factors are illustrated in the video.

- Figure 3. I am not sure how informative this plot is. If the author wants to keep it, maybe consider replacing the Spanish with English to increase the readability of the figure.

This figure has been replaced by one with English only.

- Page 18. Table without caption.

The table had slipped below to the next page, and the caption is on p17. We have shifted the whole of Table 2 to the next page, leaving some white space though at the end of the previous page. We are not sure if this will be preserved when the files are converted to PDF by the system.

- Page 24. The authors used 3 paragraphs to picture the future work that they want to conduct. I suggest replacing them with some limitations of the study.

We have extended both the limitations and future work. We feel that these are interconnected. Page 25, lines 8-26.

- Since this paper is likely of interest to a wide multi-disciplinary community (both ML and VR researchers), the authors should consider positioning the work to also be accessible to machine learning researchers. While the paper relies on a straightforward RL technique (Q-learning) and does not make a machine learning contribution, the proposed task definition may be of interest to the wider ML community. Positioning this task in relation to well-studied RL tasks would be beneficial.

We have added some information about this on page 6, lines 1-6; page 25, lines 32-34.